# Enhanced Humoral Immune Responses against Toxin A and B of *Clostridium difficile* is Associated with a Milder Disease Manifestation

**DOI:** 10.3390/jcm9103241

**Published:** 2020-10-10

**Authors:** Wasef Na’amnih, Yehuda Carmeli, Valeria Asato, Sophy Goren, Amos Adler, Dani Cohen, Khitam Muhsen

**Affiliations:** 1Department of Epidemiology and Preventive Medicine, School of Public Health, Sackler Faculty of Medicine, Tel Aviv University, Tel Aviv 6139001, Israel; valeria.asato@gmail.com (V.A.); sophyg@tauex.tau.ac.il (S.G.); amosa@tlvmc.gov.il (A.A.); dancohen@tauex.tau.ac.il (D.C.); 2Department of Geriatric Rehabilitation, Tel-Aviv Sourasky medical Center, Tel Aviv 6423906, Israel; 3Division of Epidemiology, and the National Institute for Antibiotic Resistance and Infection Control, Tel Aviv Sourasky Medical Center, Tel Aviv 6423906, Israel; yehudac@tlvmc.gov.il; 4Clinical Microbiology Laboratory, Tel-Aviv Sourasky medical Center, Tel Aviv 6423906, Israel

**Keywords:** immunoglobulin G, immunoglobulin A, *Clostridium difficile*, toxin A, toxin B, disease severity, sero-epidemiology

## Abstract

The role of the humoral immune response to *Clostridium difficile* in modulating the severity of *C. difficile* infection (CDI) is unclear. We compared the levels of serum immunoglobulin G (IgG) and immunoglobulin A (IgA) against toxin A (TcdA) and toxin B (TcdB) of *C. difficile* between CDI and control patients and according to disease severity. The levels of IgG and IgA antibodies against TcdA and TcdB were measured in sera from patients with CDI (*n* = 50; 19 had severe CDI) and control patients (*n* = 52), using ELISA. Patients with CDI had higher levels of IgG antibodies against TcdA and TcdB than controls (*p* = 0.001 and *p* = 0.04, respectively). Higher IgG levels against TcdA and TcdB were found in patients with mild vs. severe CDI 7–14 days after the diagnosis (*p* = 0.004 and 0.036, respectively). A factor analysis included both IgA and IgG levels against both toxins into one composite variable, which was of higher values in patients with mild vs. severe CDI (*p* = 0.026). In conclusion, the systemic humoral immune responses against TcdA and TcdB might modulate the severity of CDI. These preliminary findings provide a basis for future large-scale studies and support the development and evaluation of active and passive immunotherapies for CDI management.

## 1. Introduction

*Clostridium difficile* is the most common cause of infectious diarrhea in hospitalized patients [1,2]. The bacterium has two toxins, A and B, that play a pivotal role in the pathogenesis of the disease [3]. Up to 35% of patients have recurrent *C. difficile* infection (CDI) [4,5]. *C. difficile* can cause asymptomatic colonization, as well as disease with various degrees of severity from mild diarrhea to fulminant colitis and death [6,7,8]. Such variability in the outcome of *C. difficile* infection can be attributed to the patient’s characteristics, including advanced age, the severity of underlying comorbidities, and the systemic humoral immune response against *C. difficile* toxins [9,10,11,12].

Epidemiological observational studies have demonstrated the important role of the systemic humoral immune response directed against *C. difficile* toxin A (TcdA) and B (TcdB) in the risk of primary and recurrent CDI [10,11,12,13,14,15] and mortality following CDI [9]. Kyne et al. have demonstrated high levels of serum immunoglobulin G (IgG) directed against TcdA among asymptomatic carriers of *C. difficile* compared to persons who developed diarrhea [12]. They also showed an increased risk for recurrent CDI in persons with low serum IgG antibodies against TcdA compared to those with a high antibody level [11]. Solomon et al. showed that low anti-TcdA IgG titer was significantly associated with 30-day all-cause mortality in patients with severe CDI [9]. Such an association was not found for TcdB IgG antibodies [9]. Others, including recent studies, on the other hand, have highlighted the role of serum IgG or IgA against TcdB, rather than TcdA, in CDI outcome [10,13,14,15,16,17]. Recent clinical trials (MODIFY I and MODIFY II) with the human monoclonal antibodies actoxumab and bezlotoxumab against TcdA and TcdB, respectively, showed that among participants receiving antibiotic treatment for primary or recurrent CDI, bezlotoxumab was associated with a lower rate of recurrent infection than placebo, while the addition of actoxumab did not improve efficacy [18]. Secondary analysis of the placebo group in these trials showed an inverse association between high endogenous antibody titers against TcdB but not TcdA and recurrent CDI [19]. Despite this evidence, there are unresolved questions, namely the relationship of the humoral immune responses against *C. difficile* toxins with the disease severity of CDI and the relative importance of serum antibodies against toxins TcdA and TcdB. The main aim of this study was to examine the differences in serum IgG and IgA levels against TcdA and TcdB between CDI and control patients and according to CDI severity. Our hypothesis was that patients with mild CDI might have higher humoral responses compared to patients with severe CDI. The evaluation of the humoral immune responses according to the severity of CDI is novel, compared to previous studies that mostly focused on the prevention and recurrence of CDI.

## 2. Materials and Methods

### 2.1. Study Design and Population

A case-control study on the risk factors of CDI was undertaken in 2011–2014 at Tel Aviv Sourasky Medical Center in Israel. Details on the study are published elsewhere [20,21]. Briefly, hospitalized cases with CDI and hospital controls not suffering from diarrhea were recruited. Cases and controls were matched by age (±5 years), sex, hospitalization ward (medical or surgical) and number of hospitalization days (±5 days). The selection of hospital controls ensures that cases and controls were from the same source population and balances potential variation in referral patterns. The current study is based on a random subsample of 50/140 cases (35.7%) and 52/140 (37.1%) controls for whom sera were obtained. The subsample of cases and controls included in the current study was similar in terms of demographic and selected clinical characteristics compared to the entire sample (Appendix A). Information was collected from medical records on age (in years), sex, time since CDI diagnosis (defined as difference in days between the date of blood collection and laboratory diagnosis of *C. difficile*), and laboratory results of complete blood count, blood albumin and serum creatinine levels during hospitalization.

Patients with CDI were classified as having a severe disease if they had leukocytosis with a white blood cell count of ≥15,000 cells/µL, decreased blood albumin (<30 g/L) or a rise in serum creatinine level ≥1.5 times the premorbid level. Patients with CDI who did not fulfill any of these conditions were classified as having a mild disease.

### 2.2. Specimens’ Handling and Laboratory Methods

Stool specimens were collected from patients with diarrhea and from the control group and were tested for *C. difficile* using an enzyme immunoassay for the detection of toxin A/B (*C. difficile* toxin A/B II, Techlab, Blacksburg, VA) in 2011. In 2012, a 2-step algorithm was introduced. A combined glutamate dehydrogenase (GDH) and Toxin A/B immunochromatographic rapid assay (C. DIFF QUIK CHECK^®^ -Techlab, Orlando, FL, USA) was used at the initial step; if results were discordant, a polymerase chain reaction (PCR) (Xpert *C. difficile*, Cepheid, Sunnyvale, CA, USA) was utilized in the second step [20,21].

Blood samples were collected from 50 CDI cases and 52 controls. In a subsample of 20 cases, an additional blood sample was collected 14 days after the collection of the first sample to assess the change in the antibody levels.

The levels of serum IgG and IgA antibodies against TcdA and TcdB were measured using an enzyme-linked immunosorbent assay (ELISA) commercial kit, according to the manufacturer’s instructions (tgcBIOMICS, Mainz, Germany) [22]. The Optical Density (OD) was measured using an ELISA plate reader (Thermo Scientific Multiscan FC (Waltham, MA, USA)) at 450 and 620 nm; the latter was considered as the background. The antibody level is expressed in ELISA units (EUs) according to the formula: OD measurements (OD 450–620 nm) × dilution factor. If more than one dilution was performed, the average result was calculated.

### 2.3. Statistical Analysis

The geometric mean titers (GMTs) (and standard deviation (SD)) of serum IgA and IgG levels against TcdA and TcdB were calculated. Differences between cases and controls in the GMTs of IgA and IgG against TcdA and TcdB were examined using a Student’s *t* test and the differences according to CDI severity were examined using the Mann Whitney test. An additional analysis was restricted to samples collected 7–14 days after the laboratory diagnosis of *C. difficile* based on the assumption that it takes more than one week until the rise in serum antibodies. In a subsample with paired sera obtained 2 weeks apart, we examined the change in serum IgA and IgG levels against TcdA and TcdB using the paired t test. Statistical significance was set at *p* < 0.05. Adjustment for multiple comparisons was conducted using the Benjamini–Hochberg false discovery rate method [23]. We assumed that immunological markers might be correlated; therefore, we examined the correlations between IgA and IgG antibodies against TcdA and TcdB using the Spearman Rank correlation coefficient. The antibody levels were used in a factor analysis to explore data reduction and create a composite variable representing the humoral immune response [24,25] against *C. difficile*; the newly created composite variable was expressed as a Z score and differences in the this newly created composite variable according to CDI severity was examined using the Mann Whitney test. Data were analyzed using SPSS version 25 (IBM, New York, NY, USA) and Winpepi [26].

### 2.4. Ethics Statement

The study was approved by the Tel Aviv Sourasky Medical Center Ethical committee (number in the ethics committee (code) 0528-10-TLV).

## 3. Results

Overall, 50 patients (62% females) with CDI and 52 controls (56% females) were included in the study. The mean age of the CDI patients was 79.2 years (SD 14) and 82.7 years (SD 8) for the controls.

### 3.1. Differences in IgA and IgG Levels against TcdA and TcdB between CDI Cases and the Controls

Patients with CDI had significantly higher GMT values of serum IgG antibody against TcdA compared to the control group: 20.1 EU (SD 2.5) vs. 11.6 EU (SD 2.1), *p* = 0.001, and against TcdB: 18.0 EU (SD 2.6) vs. 12.0 EU (SD 2.7), *p* = 0.04. A similar trend was observed for serum IgA antibodies, but the differences were not statistically significant (Table 1).

### 3.2. Differences in IgA and IgG Levels against TcdA and TcdB between Vatients with Mild and Severe CDI

A significantly higher GMT of serum IgA antibodies against TcdB was found among CDI patients with mild disease compared to patients with severe disease 9.2 EU (SD 2.7) vs. 4.9 EU (SD 1.8), *p* = 0.023. A similar but nonstatistically significant trend was found for serum IgA and IgG levels against TcdA and IgG against TcdB. Limiting the analysis to sera that were collected at days 7–14 following the diagnosis of *C. difficile* showed significantly higher IgG levels against TcdA and TcdB in patients with mild CDI compared to patients with severe CDI (Table 2).

### 3.3. Correlations between IgG and IgA against TcdA and TcdB

Significant correlations were found between serum IgG levels against TcdA and TcdB (Spearman’s r = 0.31), IgA levels against TcdA and TcdB (Spearman’s r = 0.53) and IgG and IgA levels against TcdB (Spearman’s r = 0.43) (Figure 1). No other significant correlations were found.

### 3.4. Factor Analysis of Serum IgG and IgA against TcdA and TcdB

A factors analysis showed values of 0.5 for the Kaiser–Meyer–Olkin measure and significant (*p* < 0.001) Bartlett’s test, thus indicating that the factor analysis was appropriate. One component with eigenvalues >1 was identified and it included all markers, i.e., serum IgG and IgA levels against TcdA and TcdB, and it explained 52.1% of the total variance. This resulted in one composite variable representing the humoral immune response against *C. difficile*, expressed as a Z score. The value of the composite variable was significantly (*p* = 0.026) higher in patients with mild CDI than patients with severe CDI (Figure 2).

### 3.5. Paired Sera from CDI Patients

The analysis of paired sera obtained 2 weeks apart from 20 CDI patients showed a significant increase in IgG levels against TcdA and TcdB (Table 3).

## 4. Discussion

We found significantly higher levels of serum IgG antibodies against TcdA and TcdB in patients with CDI compared to the control group. This finding might be somewhat surprising, given the compelling evidence on the protective role of circulating IgG antibodies against *C. difficile* toxins in preventing colonization and primary and recurrent CDI [10,11,12,15,19]. However, unlike our study, previous studies [10,11,12,15,19] have assessed the levels of pre-existing serum antibodies and the subsequent outcome of *C. difficile* infection. Sera obtained from the CDI patients in our study mainly represent the early and late convalescent phase of disease that typically shows increases in serum antibodies over the acute phase levels [14,27,28,29]. Our finding of a higher systemic humoral response against *C. difficile* toxins than the control group is thus reasonable and compatible with findings from studies that employed a similar study design [28,29,30].

The main findings of this study are the differences in the humoral immune response against TcdA and TcdB in relation to CDI severity. We found a higher level of serum IgA antibodies against TcdB among CDI patients with mild disease compared to patients with severe CDI. About 7–14 days after CDI diagnosis, the levels of serum IgG levels against TcdA and TcdB were significantly higher in patients with mild vs. severe CDI. These findings suggest that serum antibodies against TcdA and TcdB likely moderate the severity of CDI. These results add new supportive evidence regarding the importance of circulating serum antibodies against both toxins in moderating the disease severity. While earlier studies have emphasized anti-TcdA serum antibodies in preventing CDI and its recurrence [11,12], later studies have highlighted the significance of antibodies directed against TcdB [10,18]. However, the dissection of the specific and independent role of antibodies against TcdA and TcdB is complex given the significant correlations between them, as evident in our findings. Another interesting point is the relative importance of serum IgA vs. IgG antibodies. *C. difficile* causes a mucosal disease with the colon being the target organ, thus the involvement of IgA, which usually correlates with mucosal immunity, seems intuitive. However, the involvement of circulating IgG antibodies in CDI severity is unique and, as mentioned above, corroborates with existing knowledge [10,11,12]. Our factor analysis supports the notion that both serum IgA and IgG levels against both TcdA and TcdB play significant roles in moderating CDI severity.

The mechanism that can explain the protective effect of serum IgA and IgG antibodies directed towards *C. difficile* toxins is still not fully clear. However, these circulating antibodies were shown to correlate with neutralizing capabilities [27], although such correlations were not always demonstrated [13,27]. Interestingly the neutralizing capabilities were demonstrated in convalescent sera [27]. This supports the observed inverse association that we report between serum IgG antibody levels and disease severity 7–14 days after the diagnosis of CDI (Table 2), which accompanies the increased IgG levels against both toxins in paired sera (Table 3). Collectively, ours and others’ findings [10,11,12] re-establish the importance of circulating serum IgG and IgA levels against both TcdA and TcdB in the prevention and control of CDI.

Our studies on other bacterial pathogens, namely *Shigella* spp., that target the colon and cause dysentery also show the importance of circulating IgG antibodies in the prevention of shigellosis [31]. The pre-existing serum IgG antibody level against *Shigella* lipopolysaccharides (LPSs) was shown to be protective against natural infection caused by *Shigella sonnei* [32]. Clinical trials with conjugate *S. sonnei* vaccines (based on the LPS linked with carrier protein) administered intramuscularly resulted in significant protection against the disease in young adults [33]. The vaccine also resulted in increases in serum IgG level against *S. sonnei* LPS [33] and it has been proposed as the correlate of protection [31]. Similarly, serum IgG antibodies against *C. difficile* toxins might serve as a correlate of protection in clinical trials with active vaccines targeting *C. difficile*. Currently there is no licensed *C. difficile*-active vaccine but several are in clinical development [34,35,36]. Based on the current evidence from observational studies [9,10,11,12,19] and our new findings, the concept of presenting antigens that can prime or boost the immune system towards the production of antitoxin circulating antibodies seems a sensible approach for developing preventive and therapeutic vaccines and technologies for CDI.

The additional implications of our findings relate the use of monoclonal antibodies for the prevention of CDI. Recent clinical trials with the human monoclonal antibodies, bezlotoxumab against TcdB, demonstrated a significant efficacy in preventing recurrent CDI, while the addition of TcdA monoclonal antibody actoxumab [18] did not improve the efficacy of preventing recurrence. Bezlotoxumab was approved for the prevention of recurrent CDI by the regulatory agencies in US and Europe [37]. Our finding suggests that antibodies against TcdA might be needed for moderating the severity of CDI. Moreover, our findings suggest the need to expand the assessment of efficacy of bezlotoxumab and actoxumab in reducing CDI severity and complications in future large-scale studies. Altogether, ours and others’ findings provide supportive evidence for accelerating the development and evaluation of active and passive immunotherapies for the management of CDI, as an alternative or supplement to existing antibiotic treatments or fecal microbiota transplants.

The strong points of this study include the well-defined prospective case-control study of CDI risk factors [20,21], in which patients’ characteristics and samples were collected according to standardized protocols, the comprehensive assessment of both serum IgA and IgG levels against both *C. difficile* toxins, and the analytical approach that signifies the importance of all these markers in CDI severity. Our study has limitations—namely, the small sample size, especially the number of participants in the subgroups of disease severity, in addition to the lack of pre-existing sera. The selection of hospital controls might be a limitation since they do not necessarily represent the general population. However, immunosuppression conditions that might negatively affect the immune system, such as cancer and chemotherapy, were more common among the CDI cases than the controls [20,21].

## 5. Conclusions

Our findings are preliminary and suggest that serum antibodies directed against both toxin A and B of *C. difficile* might play a role in modulating the severity of CDI. They also provide a basis for future large-scale studies. These results call for accelerating the development and evaluation of active and passive immunotherapies for the management of CDI.

## Figures and Tables

**Figure 1 jcm-09-03241-f001:**
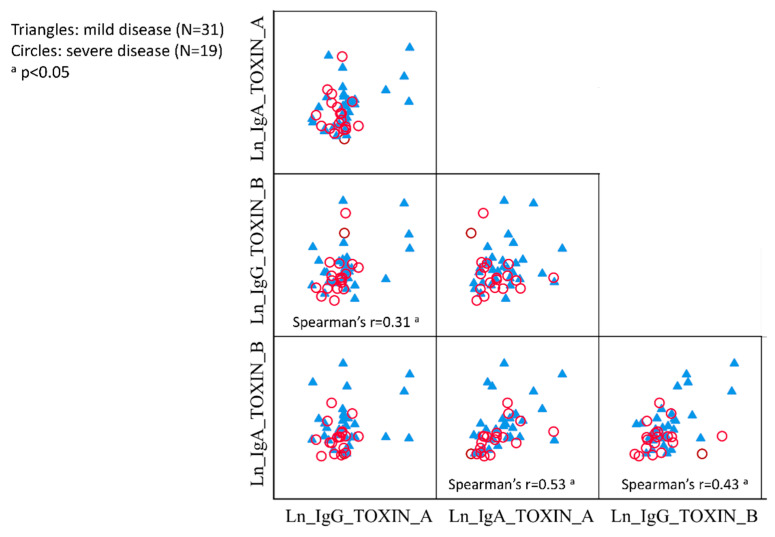
Correlation matrix between serum IgG and IgA levels against toxin A and toxin B of *Clostridium difficile* in patients with mild and severe CDI. CDI: *Clostridium difficile* infection; IgA: Immunoglobulin A; IgG: Immunoglobulin G; Ln natural logarithm.

**Figure 2 jcm-09-03241-f002:**
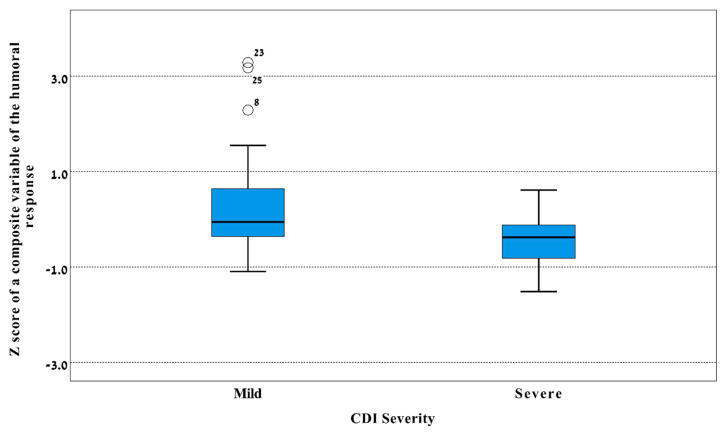
Box plot of Z score of the humoral immune response against *Clostridium difficile* in patients with mild and severe CDI. A composite variable representing the humoral immune response was created using a factor analysis and it comprises serum IgG and IgA levels against toxin A and toxin B and expressed in Z scores (Y-axis). The line in the middle of the box represents the median level, lower bound of the box represents the 25th percentile, the upper bound of the box represents the 75th percentile, the lowest point of the lower whisker represents the minimum and the highest point of the upper represents the maximum. Black circles represent outliers. *p* = 0.026 by the Mann Whitney test for the difference between sever and mild CDI (X-axis) in the median level of this variable. CDI: *Clostridium difficile* infection; IgA: Immunoglobulin A; IgG: Immunoglobulin G.

**Table 1 jcm-09-03241-t001:** Geometric mean titers (and standard deviations) of serum IgG and IgA antibody levels against toxin A and B of *Clostridium difficile* among cases and controls ^a^.

	Cases (*n* = 50) GMT (SD)	Controls (*n* = 52) GMT (SD)	*p* Value ^b^
Toxin A			
Serum IgG antibody	20.1 (2.5)	11.6 (2.1)	0.001
Serum IgA antibody	6.8 (2.1)	5.3 (2.0)	0.08
Toxin B			
Serum IgG antibody	18.0 (2.6)	12.0 (2.7)	0.04
Serum IgA antibody	7.2 (2.5)	5.4 (2.2)	0.09

^a^ The results are expressed in ELISA units; ^b^
*p* value was obtained by Student’s *t* test; GMT: geometric mean titer; IgA: Immunoglobulin A; IgG: Immunoglobulin G; SD: standard deviation.

**Table 2 jcm-09-03241-t002:** Geometric mean titers (and standard deviations) of serum IgG and IgA antibody levels against toxin A and B of *Clostridium difficile* among cases according to *Clostridium difficile* infection (CDI) disease severity.

	Toxin A						Toxin B				Toxin A	
Variable	IgG	Mean Difference (95% CI)	*p* Value ^a^	IgA	Mean Difference (95% CI)	*p* Value ^a^	IgG	Mean Difference (95% CI)	*p* Value ^a^	IgA	Mean Difference (95% CI)	*p* Value ^a^
	GMT (SD)			GMT (SD)			GMT (SD)			GMT (SD)		
**Disease severity**												
Mild (*n* = 31)	23.4 (1.1)	0.39 (−0.13, 0.92)	0.12 ^b^	7.6 (0.9)	0.30 (−0.13, 0.74)	0.14 ^c^	20.5 (1.0)	0.34 (−0.22, 0.91)	0.11 ^d^	9.2 (1.0)	0.63 (0.10, 1.15)	0.023 ^e^
Severe (*n* = 19)	15.8 (0.5)	Reference		5.6 (0.7)	Reference		14.6 (0.9)	Reference		4.9 (0.6)	Reference	
**Disease severity in the time period, 7–14 days**												
Mild (*n* = 9)	35.5 (1.1)	0.90 (0.07, 1.74)	0.004 ^f^	8.1 (0.5)	0.33 (−0.20, 0.87)	0.23 ^g^	30.1 (1.1)	0.77 (−0.40, 1.95)	0.036 ^h^	9.0 (1.2)	0.63 (−0.32, 1.57)	0.27 ^i^
Severe (*n* = 8)	14.4 (0.4)	Reference		5.8 (0.6)	Reference		13.9 (1.2)	Reference		4.8 (0.5)	Reference	

CDI: *Clostridium difficile* infection; GMT: geometric mean titer; IgA: Immunoglobulin A; IgG: Immunoglobulin G; SD: standard deviation. The results are expressed in ELISA units. ^a^
*p* value was obtained by Mann Whitney test for the difference between patients with mild vs. severe CDI. *p* value adjusted for multiple comparisons: ^b^ 0.19, ^c^ 0.14, ^d^ 0.19, ^e^ 0.09, ^f^ 0.032, ^g^ 0.26, ^h^ 0.09, ^i^ 0.27.

**Table 3 jcm-09-03241-t003:** The geometric mean titers of serum IgG and IgA antibodies level against toxin A and B of *Clostridium difficile* obtained from CDI patients 14 days apart.

	1st Sample GMT (SD)	2nd Sample GMT (SD)	*p* Value ^a^
Toxin A, IgG	15.8 (1.3)	18.9 (1.3)	0.05
Toxin A, IgA	6.9 (1.4)	7.1 (1.5)	0.8
Toxin B, IgG	17.5 (1.3)	23.8 (1.3)	0.02
Toxin B, IgA	6.1 (1.6)	6.2 (1.6)	0.9

^a^*p* value was obtained by paired *t* test. CDI: *Clostridium difficile* infection; GMT: geometric mean titer IgA: Immunoglobulin A; IgG: Immunoglobulin G; SD: standard deviation.

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
