# Peer review of "Enhanced Humoral Immune Responses against Toxin A and B of *Clostridium difficile* is Associated with a Milder Disease Manifestation"

_jcm, 2020, doi:10.3390/jcm9103241_

Round 1

Reviewer 1 Report

The manuscript describes a case control study designed to further elucidate differences in serum IgG and IgA levels against TcdA and TcdB between CDI and controls patients and according to CDI severity.

The study is succinct and concise. There are a few number of issues that would deserve attention:

- The aims of this study are defined in the last paragraph of the introduction section. Also, the authors should explain why is this contribution novel and why it is superior to the previously published studies.

- In the results section, the authors need to explain better and in more depth, the main findings (laboratory results) of the study and then present the statistical correlations. In the present form it is a bit confusing. 

Author Response

We thank the reviewers for the positive, helpful and important comments.

Reviewer 1

English language and style

 (x) English language and style are fine/minor spell check required 

Reply: We have edited the English language as suggested by the reviewer. Please see the revised manuscript.

Comments and Suggestions for Authors

The manuscript describes a case control study designed to further elucidate differences in serum IgG and IgA levels against TcdA and TcdB between CDI and controls patients and according to CDI severity.

The study is succinct and concise. There are a few number of issues that would deserve attention:

Reply: We thank the reviewer for the positive and encouraging comments.

- The aims of this study are defined in the last paragraph of the introduction section. Also, the authors should explain why is this contribution novel and why it is superior to the previously published studies.

Reply: Following the reviewer's comment, we emphasized the novelty of our study and the new evidence gained from it over previous studies. The following sentence was added to the introduction: "The evaluation of the humoral immune responses according to the severity of CDI is novel, compared to previous studies that mostly focused on prevention and recurrence of CDI." Please see the introduction section page 2 lines, 22-24.      

- In the results section, the authors need to explain better and in more depth, the main findings (laboratory results) of the study and then present the statistical correlations. In the present form it is a bit confusing. 

Reply: Following the reviewer's comment, we have edited the results section including in the text, tables and figures, to explain and simplify the results. We also added sub-heading in order to improve the flow of the results. Please the revised results section and tables/figures

Reviewer 2 Report

This is a good pilot study that is  quite underpowered and so is a good justification for a larger study.  Conclusions must be directed towards this rather than stating the certainty of their findings.  

Also the control group needs to be better defined.  A hospital control group is never normal.  It would not be difficult to include this in the text of the paper rather than just a citation, especially since it is also uncertain how this cases chosen from the original study are representative of the original whole or not.

Author Response

English language and style

(x) English language and style are fine/minor spell check required 

Reply: We have edited the English language as suggested by the reviewer. Please see the revised manuscript.

Yes

Can be improved

Must be improved

Not applicable

Does the introduction provide sufficient background and include all relevant references?

(x)

( )

( )

( )

Is the research design appropriate?

( )

(x)

( )

( )

Are the methods adequately described?

( )

(x)

( )

( )

Are the results clearly presented?

(x)

( )

( )

( )

Are the conclusions supported by the results?

( )

(x)

( )

( )

Comments and Suggestions for Authors

This is a good pilot study that is quite underpowered and so is a good justification for a larger study. Conclusions must be directed towards this rather than stating the certainty of their findings.  

Reply: We thank the reviewer for the positive and encouraging comments.  Following the reviewer's comment and suggestion, we added to the study limitations the small sample size of the study (page 9 line 18), we moderated the conclusions of the study and highlighted the preliminary nature of our findings and that can serve a basis for larger studies. Please see the conclusions section page 9 line 24-25 and the revised abstract.  

Also the control group needs to be better defined.  A hospital control group is never normal.  It would not be difficult to include this in the text of the paper rather than just a citation, especially since it is also uncertain how this cases chosen from the original study are representative of the original whole or not.

Reply: The primary aim of the original study was to assess the risk factors for C. difficile infection (CDI), in case-control study. Accordingly, the inclusion criteria for the control group was not suffering from acute diarrhea. The original case-control study included hospital controls. We agree with the reviewer that usually, hospital controls may not represent well the general population. However, in the circumstances of CDI, which is a main healthcare associated infection and major health problem in hospital setting, the selection of hospital controls is reasonable, since cases and controls come from the same source population and balances possible differences in referral patterns. This strategy also ensures that the magnitude of associations between the various risk factors and CDI status is not exaggerated as if would be expected in healthy community controls were chosen. In the original case-control study cases and controls were matched by age (±5 years), sex, hospitalization ward (surgical or medical) and number of hospitalization days (±5 days). Following the reviewer's comment, we added this information to the methods section. Please see page 2 lines 27, 29-32. The current sub-study assessed the humoral immune response to C. difficile toxins. A concern might be raised regarding the presence of immunosuppression conditions in the control group that might negatively affect the immune system However, as expected, conditions such as cancer and chemotherapy were more common among the cases than the controls 1,2 Please see the discussion section page 9 line 19-22

The patients included in the current study were randomly selected from the original study. The sub-sample of cases included in the current study had comparable characteristics to the entire cohort. Please see the table below. This information was added to text; please see page 2 line 34-36 and the table was added as a supplementary table to the manuscript.

Table 1: Demographic and selected clinical characteristics of cases and control of the original study and the sub-sample included in the current study

Variable

All cases, N=140

Sub-sample of cases, N=50

All controls, N=140

Sub-sample of controls, N=52

Age, years, Mean (SD)

78.8 (15.4)

79.2 (13.7)

81.4) 7.5)

82.7 (7.6)

Sex (Female), N (%)

88 (63%)

31 (62%)

86 (61%)

29 (56%)

Visit to emergency department in the previous year (Yes), N (%)

36 (31%)

12 (24%)

43 (31%)

20 (39%)

Antibiotic use (Yes), N (%)

117 (84%)

37 (74%)

68 (49%)

25 (48%)

Use of proton pump inhibitors (Yes), N (%)

98 (70%)

31 (62%)

101 (72%)

37 (71%)

     References  

  1. Na'amnih W, Adler A, Miller-Roll T, Cohen D, Carmeli Y. Incidence and Risk Factors for Community and Hospital Acquisition of Clostridium difficile Infection in the Tel Aviv Sourasky Medical Center. Infection control and hospital epidemiology. 2017;38(8):912-920.
  2. Muhsen K, Na'amnih W, Adler A, Carmeli Y, Cohen D. Clostridium difficile-associated disease and Helicobacter pylori seroprevalence: A case-control study. Helicobacter. 2020;25(1):e12668.

Round 2

Reviewer 1 Report

All comments have been addressed  in the revised ms.